# Profiling the Spatial Expression Pattern and ceRNA Network of lncRNA, miRNA, and mRNA Associated with the Development of Intermuscular Bones in Zebrafish

**DOI:** 10.3390/biology12010075

**Published:** 2022-12-31

**Authors:** Weidong Ye, Mijuan Shi, Keyi Ren, Yuhang Liu, You Duan, Yingyin Cheng, Wanting Zhang, Xiao-Qin Xia

**Affiliations:** 1Institute of Hydrobiology, Chinese Academy of Sciences, Wuhan 430072, China; 2College of Advanced Agricultural Sciences, University of Chinese Academy of Sciences, Beijing 100049, China; 3The Innovative Academy of Seed Design, Chinese Academy of Sciences, Beijing 100101, China; 4College of Fisheries and Life Science, Dalian Ocean University, Dalian 116023, China

**Keywords:** intramuscular bones, ceRNA network, miRNA, lncRNA, zebrafish

## Abstract

**Simple Summary:**

Worldwide, many of the major economic fish have intermuscular bones (IBs). The existence of IBs leads to potential edible risks, lowers the quality of fish, and is not conducive for further processing and expanding the consumer market. The development of IBs is regulated by complex molecular networks, including mRNA–miRNA–lncRNA ceRNA networks. Using zebrafish as a model, this study revealed, for the first time, the spatial expression pattern and ceRNA networks of lncRNA, miRNA, and mRNA associated with IB development. In addition, we identified a series of key genes and pathways that may be involved in the formation and growth of IBs. Our findings provide new insights into the molecular mechanisms of IB development and clues for improving IB traits.

**Abstract:**

Intermuscular bones (IBs) are small spicule-like bones in the muscular septum of fish, which affect their edible and economic value. The molecular mechanism of IB development is still uncertain. Numerous studies have shown that the ceRNA network, which is composed of mRNA, lncRNA, and miRNA, plays an important regulatory role in bone development. In this study, we compared the mRNA, lncRNA, and miRNA expression profiles in different IB development segments of zebrafish. The development of IBs includes two main processes, which are formation and growth. A series of genes implicated in the formation and growth of IBs were identified through gene differential expression analysis and expression pattern analysis. Functional enrichment analysis showed that the functions of genes implicated in the regulation of the formation and growth of IBs were quite different. Ribosome and oxidative phosphorylation signaling pathways were significantly enriched during the formation of IBs, suggesting that many proteins are required to form IBs. Several pathways known to be associated with bone development have been shown to play an important role in the growth of IBs, including calcium, ECM-receptor interaction, Wnt, TGF-β, and hedgehog signaling pathways. According to the targeting relationship and expression correlation of mRNA, lncRNA, and miRNA, the ceRNA networks associated with the growth of IBs were constructed, which comprised 33 mRNAs, 9 lncRNAs, and 7 miRNAs. This study provides new insight into the molecular mechanism of the development of IBs.

## 1. Introduction

Intermuscular bones (IBs) are small spicule-like bones formed due to the ossification of the tendon in the muscular septum of fish through intramembranous osteogenesis [1]. Many of the world’s major economic fish have IBs, including Cypriniformes, Clupeiformes, and Salmoniformes [2]. The existence of IBs leads to potential edible risks and lowers the quality of fish, which is not conducive for further processing and expanding the consumer market. Previous studies obtained fish with or without fewer IBs from nature or through molecular biological means [3,4,5,6], indicating the feasibility of the genetic improvement of IB traits. Some genes associated with the development of IBs have been identified via gene knockouts, such as *scxa*, *runx2b* and *sp7* [6,7,8]. IB research has progressed from the early morphology to the study of the molecular mechanism of IBs. In recent years, omics technologies have also been applied to study the molecular mechanism of IBs, including genome, transcriptome, proteome, and multi-omics integrations [2,9]. These studies have significantly improved our understanding of the molecular mechanism of the development of IBs. However, the exact molecular mechanism remains unknown.

Bone development is a complex process regulated by multilayer and multinetwork factors. Numerous studies have shown that bone development is also regulated by noncoding RNAs, such as microRNA and lncRNA [10,11]. However, there are few studies on the regulation of the development of IBs by noncoding RNAs, which have only been reported in common carp and blunt snout bream [12,13,14]. Salmena proposed the competitive endogenous RNA (ceRNA) hypothesis, which significantly influences the regulation patterns of mRNA, lncRNA, and microRNA [15]. This hypothesis proposes that mRNA, lncRNA, and other noncoding RNAs can communicate and regulate each other’s expression levels by competitively binding to the miRNAs that participate in cell activity and function regulation. ceRNA networks have been applied to study many complex traits, such as bone development, growth, and disease resistance [16,17,18]. Chen Yulong et al. constructed a ceRNA network related to the development of IBs of blunt snout bream by comparing the mRNA, lncRNA, and miRNA expression profiles of IBs at the age of 1 and 3 [12].

The zebrafish is an important model organism and a Cyprinidae fish with IBs. It has been used to study the molecular mechanism of the development of IBs. Like other Cyprinidae fishes, the IBs of zebrafish gradually form from tail to head [19]. The IBs in living zebrafish can be continuously tracked by calcein staining [20]. The zebrafish IBs are formed starting in the tail at 20 dpf (body length 6.5 mm) and grow to the mid-lower dorsal fin at 30 dpf (15 mm). The development of IBs cannot be accurately observed after 30 dpf due to the shielding of scales and body surface pigments. In this study, the mRNA, lncRNA, and miRNA expression profiles of three IB segments (IBs have not yet formed, few IBs have been formed, and all IBs have been formed) were obtained through the whole transcriptome sequencing in 30 dpf zebrafish. Based on differentially expressed genes, mRNAs, lncRNAs, and miRNAs associated with the development of IBs were obtained through gene expression pattern analysis, and the mRNA–lncRNA–miRNA ceRNA networks associated with IB development were constructed. This study provides a deeper understanding of the key genes associated with IB development and advances our understanding of the molecular mechanism of IB development.

## 2. Materials and Methods

### 2.1. Experimental Fish

The wild-type zebrafish used in the experiment were of the AB strain, and were purchased from the China Zebrafish Resource Center. Embryos and juveniles (≤30 dpf) were cultured in a constant-temperature and light incubator, whereas juveniles (>30 dpf) and adults were cultured in a circulating water system. The breeding water temperature was controlled at 28 ± 0.5 °C, and the illumination time was 14 h per day.

### 2.2. Sample Collection and RNA Extraction

Calcein staining was used to observe the development of IBs in zebrafish. The 30 dpf zebrafish with IBs that grow to the mid-lower dorsal fin were used for sampling, including the S1 segment (IBs have not yet formed), the S2 segment (few IBs have been formed), and the S3 segment (all IBs have been formed) (Figure 1). Zebrafish from the same family and at the same developmental state were selected for sampling. There was no interval between the three segments sampled, and the S2 segment contained IBs undergoing the process of formation. After anesthesia with 50 mg/L Tricaine methanesulfonate (MS-222), muscle tissue containing only IBs (i.e., excluding skin, vertebra, blood, and excess muscle) was collected under an anatomical microscope. The tissue was added to a 1.5 mL centrifuge tube containing 1 mL of TRIzol and 20 grinding beads, and was ground at 60 Hz for 2 min in a cryogenic grinder (JXFSTPRP24, Shanghai Jingxin Industrial Development Co., Ltd, Shanghai, China) at 4 °C. Total RNA was isolated from tissue suspensions using the TRIzol reagent (Thermo Fisher Scientific, San Jose, CA, USA). According to the total RNA quality, 3 total RNA samples from the same segment of different individuals were selected and mixed for sequencing, with 3 biological replicates used for each segment.

### 2.3. Library Construction and Sequencing

For miRNA, the library was constructed using a small RNA sample prep kit. First, the total RNA was taken as the starting sample by directly adding connectors to both ends of the small RNA (the complete phosphate group at the 5’ end and the hydroxyl group at the 3’ end), and then reverse transcribing to synthesize cDNA using the special structure of 3’and 5’ ends of small RNA (the complete phosphate group at the 5’ end and the hydroxyl group at the 3’ end). Then, after PCR amplification, the target DNA fragment was separated via PAGE gel electrophoresis, and the cDNA library was recovered by cutting the gel. After construction, the library’s quality was evaluated using the Agilent Bioanalyzer 2100 system. After passing the quality control checks, libraries were barcoded and multiplexed for sequencing on the Illumina HiSeq 2500 platform in a 50 bp, single-end format.

The total RNA was isolated using the TRIzol Reagent (Thermo Fisher Scientific, San Jose, CA, USA), after which the concentration, quality, and integrity were determined using a NanoDrop spectrophotometer (NanoDrop Technologies, Wilmington, DE, USA). After the rRNA was removed with the Epicentre Ribo-Zero^TM^ rRNA Removal Kit (Epicentre, Madison, WI, USA), sequencing libraries were constructed using the NEBNext^R^ Ultra^TM^ Directional RNA Library Prep Kit for Illumina^R^ (NEB, Ipswich, MA, USA), following the manufacturer’s instructions. PCR products were purified (AMPure XP system) and library quality was assessed using the Agilent Bioanalyzer 2100. After passing the quality control checks, libraries were sequenced on an Illumina HiSeq X Ten platform in a 150 bp, paired-end format. 

### 2.4. Assembly and Annotation of Transcriptomes

After the lncRNA clean data were mapped to the reference genome by using HISAT2 (version 2.1.0) [21,22], StringTie (Version V1.3.1c) [23] was used for transcription assembly. Salmon (version 0.12.0) [24] was used to calculate the read counts. Using CPAT (version 1.2.2) [25] and CPC2 (version 0.1) [26], all transcripts were divided into lncRNA and mRNA [27]. When the parameters used were not listed, default settings were used. 

The clean data of miRNA were aligned to the reference genome by using Bowtie (version 1.2) [28]. Additionally, reads shorter than 18 nt and longer than 31 nt were removed. The aligned reads were aligned to the Rfam (version 13.0) database [29] and the Bowtie software was used to remove zebrafish repeat sequences and ncRNAs. miRNA reference sequences were derived from miRBase (version 20.0) [30]. Finally, the mirDeep2 (version 0.1.2) software [31] was used to detect known miRNAs, predict novel miRNAs, and count miRNA expression levels.

### 2.5. Differential Expression, Gene Expression Pattern, and Function Enrichment Analysis

Differential expression analysis was performed using R Package DESeq2 (version 1.30.1) [32], and differential expression genes (DEGs) were obtained (*p* < 0.05). The STEM software (version 1.3.13) was used to analyze the expression pattern of DEGs to obtain DEGs with the same expression pattern that was associated with the development of IBs [33]. We took the mean expression of three samples at each segment for the trend analysis. R package clusterProfiler (version 4.0) was used for the GO and KEGG enrichment analysis [34], and *p* < 0.05 was considered a significant enrichment. The functional enrichment of miRNA and lncRNA is actually the functional enrichment of their target genes. The target genes of miRNA were represented by the mRNAs that have a targeting relationship with them, and their expressions were highly correlated (*r* ≥ 0.7, *p* < 0.05). The target genes of lncRNA were shown as mRNAs that had a high correlation (*r* ≥ 0.7, *p* < 0.05) with lncRNA expression.

### 2.6. Identification of Genes Associated with the Development of IBs and Construction of ceRNA Networks

We classified the DEGs associated with the development of IBs into two types based on the results of the STEM software analysis. One type was the DEGs whose expression levels showed gradient changes in the S1, S2, and S3 segments, which were possibly associated with the growth of IBs. The other is the DEGs with the highest or lowest expression levels in the S2 segment, which may be associated with the formation of IBs. The construction of the ceRNA network was based on DEGs associated with the growth of IBs. First, the target mRNA and lncRNA of miRNA were predicted using Miranda (version 3.3a) [35]. The mRNAs and lncRNAs targeted by the same miRNA were paired as mRNA–lncRNA pairs. Gene expression correlations between miRNA, mRNA, and lncRNA were measured by obtaining Spearman’s correlation coefficient. Based on the ceRNA principle, negatively correlated mRNA–miRNA pairs (*r* ≤ −0.7, *p* < 0.05) and lncRNA–miRNA pairs (*r* ≤ −0.7, *p* < 0.05) and positively correlated mRNA–lncRNA pairs (*r* ≥ 0.7, *p* < 0.05) were selected. These pairs were combined to construct mRNA–miRNA–lncRNA ceRNA networks, and the Cytoscape software (version 3.7.2) [36] was used for visualization of ceRNA networks. The online tool STRING [37] was employed to construct protein–protein interaction (PPI) networks, and interactions with a combined score bigger than 0.4 were selected.

### 2.7. Verification of Gene Expression Using RT-qPCR

Nine DEGs with gradient changes in the expression levels were selected to verify the sequencing data, including three mRNAs, three lncRNAs, and three miRNAs. Random primers were used to reverse transcribe mRNA and lncRNA into cDNA, and stem ring primers were used to reverse transcribe miRNA into cDNA. The RT-qPCR reaction was conducted using the ChamQ Universal SYBR qPCR Master Mix (Vazyme, Nanjing, China) in the CFX96 Touch Real-Time PCR (Bio-Rad Laboratories, Hercules, CA, USA). The reference gene for mRNA and lncRNA was *ef-1α*. The reference gene for miRNA was *u6* snRNA. The relative quantification of the target to the reference was performed using the 2^−ΔΔCT^ method. The primers for this study are shown in Appendix A. 

## 3. Results

### 3.1. Transcriptome Assembly and Statistics

A total of nine mRNA/lncRNA sequencing data and nine miRNA sequencing data were obtained. mRNA/lncRNA library sequencing yielded 107.81 Gb of clean reads, with an average of 11.98 Gb of clean reads per sample and an average mapping rate of 90.46% (Appendix A). miRNA library sequencing yielded 13.70 Gb of clean reads, with an average of 1.52 Gb of clean reads per sample and an average mapping rate of 98.00% (Appendix A). A total of 17,507 mRNA genes, 7154 lncRNA genes, and 395 miRNA genes were obtained from all samples (Appendix A).

### 3.2. Identification of DEGs and Functional Enrichment Analysis

The gene expression profiles between S1, S2, and S3 (S1_vs_S2, S1_vs_S3, and S2_vs_S3) were analyzed for differential expression. The results showed that S1_vs_S2 had the fewest differential expression mRNAs (DEmRNAs), differential expression lncRNAs (DElncRNAs), and differential expression miRNAs (DEmiRNAs), and S1_vs_S3 had the largest number of DEmRNAs and DElncRNAs (Figure 2A and Appendix A). S1_vs_S3 and S2_vs_S3 had the largest number of overlapping DEGs, and S1_vs_S2 and S2_vs_S3 had the least number of overlapping DEGs (Figure 2B). In addition, 42 DEmRNAs, 5 DElncRNAs, and 6 DEmiRNAs exist in S1_vs_S2, S1_vs_S3, and S2_vs_S3 (Figure 2B).

The functional enrichment analysis was performed on DEmRNAs between the S1, S2, and S3 segments. DEmRNAs of S1_vs_S2 were significantly enriched in 300 GO terms, which are mainly involved in energy metabolism, nutrient metabolism, muscle development, and bone formation systems. DEmRNAs of S2_vs_S3 were significantly enriched in 292 GO terms, which are mainly involved in translation, muscle development, vascular morphogenesis, and bone development. DEmRNAs of S1_vs_S3 were significantly enriched to 470 GO terms, which are mainly involved in energy metabolism, nutrient metabolism, muscle development, vascular morphogenesis, metal ion homeostasis, connective tissue development, and bone development. The results of the KEGG enrichment analysis showed that DEmRNAs of S1_vs_S2, S2_vs_S3, and S1_vs_S3 are all involved in nutrient metabolism, ECM–receptor interactions, and cardiac muscle contraction (Figure 2C), whereas the DEmRNAs of S1_vs_S2 and S1_vs_S3 are specifically involved in energy metabolism. In addition, the DEmRNAs of S2_vs_S3 and S1_vs_S3 are specifically involved in focal adhesion, and the DEmRNAs of S1_vs_S3 are specifically involved in the calcium signaling pathway. In general, DEmRNAs between different segments are involved in muscle development, ECM–receptor interactions, nutrient metabolism, and bone development, and are also specifically involved in energy metabolism, translation, angiogenesis, metal ion homeostasis, calcium signaling, and connective tissue development. The full functional enrichment results of DEmRNAs are presented in Appendix A.

### 3.3. Identification of mRNA, lncRNA, and miRNA Associated with IB Development

By analyzing the expression patterns of DEmRNAs, DElncRNAs, and DEmiRNAs, 16 gene expression patterns were identified in each RNA (Appendix A). We refer to the three gene expression patterns (1, 2, and 3) with a gradually increasing expression as the upregulated expression pattern (S1 < S2 < S3) (Figure 3A), and the three gene expression patterns (4, 5, and 6) with a gradually decreasing expression as the downregulated expression pattern (S1 > S2 > S3) (Figure 3A). A total of 303, 69, and 9 upregulated genes and 227, 49, and 8 downregulated genes were obtained from the DEmRNAs, DElncRNAs, and DEmiRNAs, respectively (Figure 3A and Appendix A). Considering that the development degree of IBs in the three segments changes in a gradient, we speculate that these genes with gradually increasing or decreasing expression levels may be associated with the growth of IBs. To further understand the functions of these genes, the DEmRNAs in them were subjected to a KEGG pathway enrichment analysis. These DEmRNAs were found to be involved in several known pathways associated with bone development, including calcium, ECM-receptor interaction, Wnt, TGF-β, and hedgehog signaling pathways (Table 1 and Figure 3B), which confirmed the rationality of our speculation. In addition, these genes were also significantly enriched in the functions of focal adhesion, cardiac muscle contraction, actin cytoskeleton regulation, and glucose metabolism. We also performed the functional enrichment analysis of DElncRNAs and DEmiRNAs associated with the growth of IBs, and found that their pathways are similar to those of DEmRNAs (Appendix A).

The S2 segment is the transition segment for the development of IBs, which contains IBs that are in the process of forming. We speculate that some genes that may play a crucial role in the formation of IBs have the highest or lowest expression in this segment. Therefore, DEmRNAs, DElncRNAs, and DEmiRNAs with relatively high expression patterns (7, 8, and 9) and relatively low expression patterns (10, 11, and 12) in the S2 segment were extracted (Figure 3C), which were called upregulated genes and downregulated genes, respectively. A total of 269, 99, and 20 upregulated genes and 95, 14, and 20 downregulated genes were obtained in DEmRNAs, DElncRNAs, and DEmiRNAs, respectively ((Figure 3C and Appendix A). Functional enrichment analysis of these DEmRNAs showed that ribosomal and oxidative phosphorylation signaling pathways were prominent (Figure 3D), and almost all DEmRNAs in these pathways were shown to have the highest expression levels in tS2 segment. The functional enrichment analysis of DElncRNAs and DEmiRNAs also showed similar results (Appendix A). In addition, the known pathways associated with bone development were not significantly enriched here, and only a few genes were annotated in these pathways, such as *thbs1b* in the TGF-β signaling pathway (Appendix A).

### 3.4. Construction of ceRNA Networks Associated with IB Growth

In the present study, we successfully constructed the ceRNA networks associated with the growth of IBs. The construction of the ceRNA networks was based on the genes associated with the growth of IBs, consisting of 33 mRNAs (26 upregulated and 7 downregulated), 9 lncRNAs (7 upregulated and 2 downregulated), and 7 miRNAs (1 upregulated and 6 downregulated) (Figure 4A). This network consists of six subnetworks: five upregulated (including only upregulated mRNA) subnetworks and one downregulated (including only downregulated mRNA) subnetwork. We found that the subnetwork with dre-miR-10c-5p, dre-miR-7b, MSTRG.20610, and MSTRG.23937 as the hub nodes contains the largest number of genes. There are five genes encoding collagen in this network, and all are upregulated genes, including *col5a2a*, *col15a1a*, *col16a1*, *col27a1b*, and *col6a2*. PPI analysis showed that these collagen proteins interacted with *postnb* (which encodes periostin) and *sparc* (which encodes osteonectin) from the subnetwork with dre-miR-133b-5p and MSTRG.12206 as the hub nodes, and *col5a2a* might play a central role (Figure 4B). In addition, *col5a2a* was the only gene regulated by two miRNAs (dre-miR-10c-5p and dre-miR-7b) and two lncRNAs (MSTRG.20610 and MSTRG.23937). The subnetwork with dre-miR-152, MSTRG.29519, and MSTRG.19062 as the hub nodes was the only downregulated subnetwork, and *fstl3* and *hoxa1a* in this network interacted with genes from other upregulated subnetworks, including *ctnnl1* and *srpx*.

### 3.5. Validation of RNA-Seq Data via RT-qPCR

We verified the relative expression levels of three mRNAs (*igfn1.3*, *tnni1al*, and *mb*), three lncRNAs (MSTRG.1095, MSTRG.25211, and MSTRG.32095), and three miRNAs (dre-miR-99, dre-miR-192, and dre-miR-10c-5p). The results showed that the expression trend of these genes’ RT-qPCR and RNA-seq data was consistent (Figure 5), indicating that the RNA-seq data were credible.

## 4. Discussion

Animal bones are formed through two osteogenic mechanisms, endomembranous and endochondral osteogenesis [38]. As a unique fishbone, IBs have been proven to be formed through the osteogenic mechanism of intramembrane osteogenesis [9]. The development of fish is not strictly chronological, but rather, is affected by many factors [39], as is IB development. Therefore, some methods are needed to track the development of IBs. In previous studies, IBs have been observed with X-ray, alizarin red, calcein, and Masson trichrome staining [9]. Among these methods, only calcein staining can ensure the survival of experimental fish after staining. Through calcein staining, we could ensure that the IBs of each fish were at the same stage of development before sampling, rather than relying on past inferences. This is very important for the microsampling of IBs of zebrafish. Through continuous staining observations of the same batch of zebrafish, we obtained zebrafish with basically the same developmental position of the IBs and used them for sampling. It is impossible to obtain tissues from the same part of the same zebrafish at different developmental stages because sampling will result in fish death. Furthermore, different individuals are likely to have unreliable transcriptome results due to their genetic differences. To circumvent these problems, this study first compared the differences in three IB development segments in a zebrafish by using spatial differences to reflect the temporal differences in the development of IBs. Finally, based on these samples, the expression profiles of mRNA, lncRNA, and miRNA in different IB development segments of zebrafish were analyzed for the first time in this study.

Zebrafish IBs form gradually from the tail to the head and are arranged relatively orderly in the muscles. This highly regular formation makes IBs very special as compared to other bones. In this study, according to the changes in gene expression patterns in S1, S2, and S3, we focused on two types of genes. The first type of genes only showed the highest/lowest expression levels in the S2 segment. Since the S2 segment presents IBs undergoing to formation process, we believe these genes are probably associated with IB formation and are the genes that regulate the early development of IBs. The second type of genes was continuously upregulated or downregulated in the S1, S2, and S3 segments, which is consistent with the growth trend of IBs. Therefore, we believe these genes are associated with IB growth and regulate the later development of IBs.

The enrichment analysis showed that the ribosome and oxidative phosphorylation signaling pathways were significantly enriched in the first type of genes, and almost all DEGs in these pathways were shown to have the highest expression levels in the S2 segment, indicating that a large number of proteins may be required as much energy is consumes during IB formation. Glucose metabolism signaling pathways were also enriched, suggesting that glucose metabolism may mainly supply these energies. In addition, in the TGF-β signaling pathway associated with bone development, only *thbs1b* upstream of this pathway was differentially expressed and downregulated in the S2 segment. The THBS1 protein is located in the extracellular matrix and is an important component of the bone matrix. Previous studies have revealed that the knockout of this gene affects the development of craniofacial bone in mice and leads to a decrease in the number of osteoclasts [40]. The downregulation of *thbs1b* may be closely associated with IB formation. The second type of genes associated with the growth of IBs was also enriched in the TGF-β signaling pathway. Interestingly, the genes associated with osteoblasts in this pathway were upregulated, such as *inhbaa* and *bmpr1bb* [41,42]; the genes associated with osteoclasts were downregulated, such as *smad3b* [43]. It is well known that bone homeostasis is maintained by balancing the bone formation by osteoblasts and bone resorption by osteoclasts. During the stage of rapid bone development, bone formation is greater than bone resorption [44], and IB development may follow the same principle.

The transmembrane transport of calcium ions, an important fundamental component, must have changed during the growth of IBs. The cardiac muscle contraction signaling pathway was significantly enriched in IB formation and growth, including the calcium ion transmembrane transport genes and downstream muscle-contraction-related genes. The genes associated with calcium ion transport across membranes in this pathway showed inconsistent expression patterns, suggesting the complexity of calcium ion transport regulation in the growth of IBs, including *cacna2d2a*, *cacna1sa,* and *slc8a4b*. There were also differences in genes associated with muscle contraction, suggesting that muscle contraction may also be involved in the development of IBs. Some studies have revealed that the swimming pattern of fish determines the ossification pattern of IBs [45,46], which also reflects this correlation.

The second type of genes associated with IB growth is enriched in several known pathways associated with bone development, including calcium, ECM–receptor interaction, Wnt, TGF-β, and hedgehog signaling pathways, in addition to some other metabolic pathways, all of which have been reported in previous studies [9,14,47]. These results suggest that these pathways may play specific roles in the development of IBs. In addition, some pathways related to cell migration were enriched, such as focal adhesion and actin cytoskeleton regulation pathways [48]. During the process of intramembrane osteogenesis, the migration of mesenchymal cells and preosteoblasts to proper positions is a key step, and the abnormal migration of these cells will lead to a homeostasis imbalance of bone [49]. Therefore, regulating the migration of these cells to the appropriate location may be a key step in the regular growth of IBs.

In this study, we also constructed mRNA–miRNA–lncRNA ceRNA networks for the growth of IBs in zebrafish (Figure 4A), which may regulate multiple processes of the development of IBs. In these ceRNA networks, dre-miR-10c-5p, dre-miR-7, dre-miR-133b-5p, and dre-miR-152 may play a central role, targeting 26 mRNAs and 5 lncRNAs. Dre-miR-10c-5p and dre-miR-7 were in a ceRNA subnetwork, and previous studies have shown that they are both negative regulators of osteoblast differentiation [50,51,52]. Dre-miR-10c-5p had the largest number of target genes and regulated four collagen genes, including *col5a2a*, *col15a1a*, *col16a1*, and *col27a1b*. Collagen is well known to be an important component of the bone matrix [53,54,55,56,57]. In addition, the PPI analysis showed that these collagens interacted with *sparc* (which encodes osteonectin) and *postnb* (which encodes periostin), further highlighting their role in the growth of IBs. *Sparc* encodes a secreted acidic protein and rich cysteine, also known as osteonectin, that enhances the mineralization of the collagen matrix in bone [58]. *postnb* encodes periostin, also known as osteoblast-specific factor b, which is secreted by osteoblasts and their progenitors and plays an important role in osteoblast proliferation, mineralization, and migration [59,60]. Interestingly, *postnb* and *sparc* are target genes of dre-miR-133b-5p, suggesting that these ceRNA subnetworks associated with the growth of IBs are linked rather than acting independently. In addition, the downregulated ceRNA subnetwork with dre-miR-152 as the hub node may be involved in the regulation of osteoclast differentiation. It has been reported that Tspan5 is upregulated during osteoclast differentiation and promotes RANKL-signaling-induced osteoclast formation, whereas the knockdown of Tspan5 significantly inhibits osteoclast differentiation [61,62]. In this study, *tspan5a* was downregulated during the growth of IBs and was targeted by dre-miR-152. The lncRNAs, MSTRG.29519 and MSTRG.19062, targeted by dre-miR-152 might have acted as molecular sponges of dre-miR-152 to weaken its negative regulation of *tspan5a*.

## 5. Conclusions

In conclusion, we profiled the expression profiles of mRNA, lncRNA, and miRNA in different IB development segments of zebrafish and identified a series of genes associated with the formation and growth of IBs. Ribosome and oxidative phosphorylation signaling pathways are highly enriched during the formation of IBs. Several pathways associated with bone development play important roles in the growth of IBs. The ceRNA networks associated with the growth of IBs may be involved in multiple processes regulating the growth of IBs, and some genes may play key roles in these processes, such as dre-miR-10c-5p, MSTRG.12206, *col5a2a*, *postnb*, and *sparc*. This study adds to our understanding of the key genes associated with the development of IBs and advances our understanding of the molecular mechanism of IB development.

## Figures and Tables

**Figure 1 biology-12-00075-f001:**
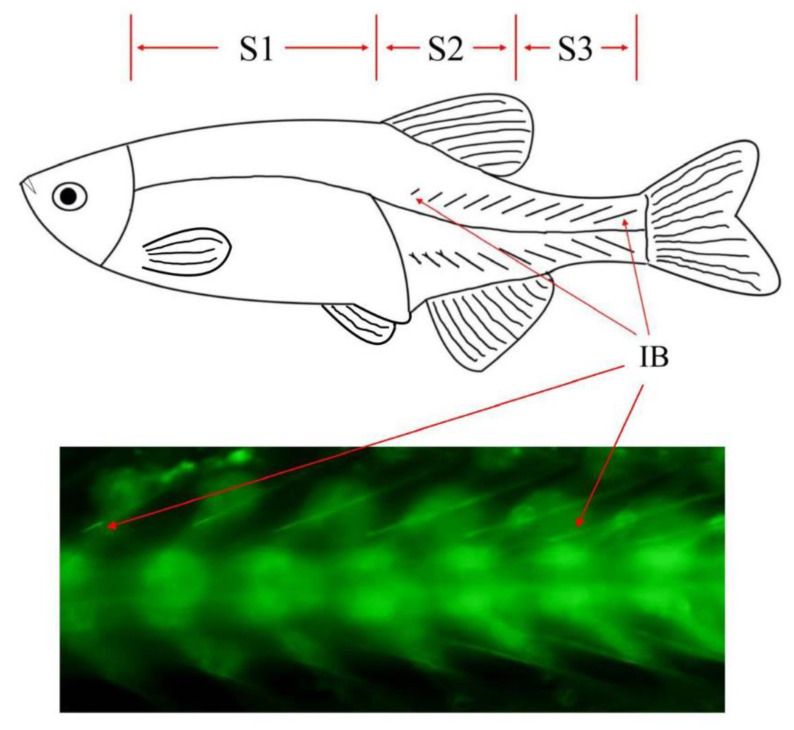
Phenotypic characteristics and sampling scheme of zebrafish IBs.

**Figure 2 biology-12-00075-f002:**
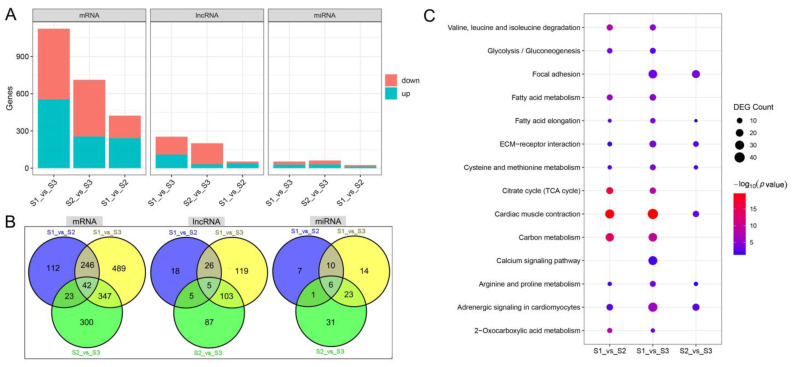
DEGs between S1, S2, and S3 segments. (**A**) The number of DEmRNAs, DElncRNAs, and DEmiRNAs of S1_vs_S2, S1_vs_S3, and S2_vs_S3. (**B**) Differences in the composition of DEmRNAs, DElncRNAs, and DEmiRNAs in S1_vs_S2, S1_vs_S3, and S2_vs_S3. (**C**) Differences in KEGG enrichment results of DEmRNAs of S1_vs_S2, S1_vs_S3, and S2_vs_S3.

**Figure 3 biology-12-00075-f003:**
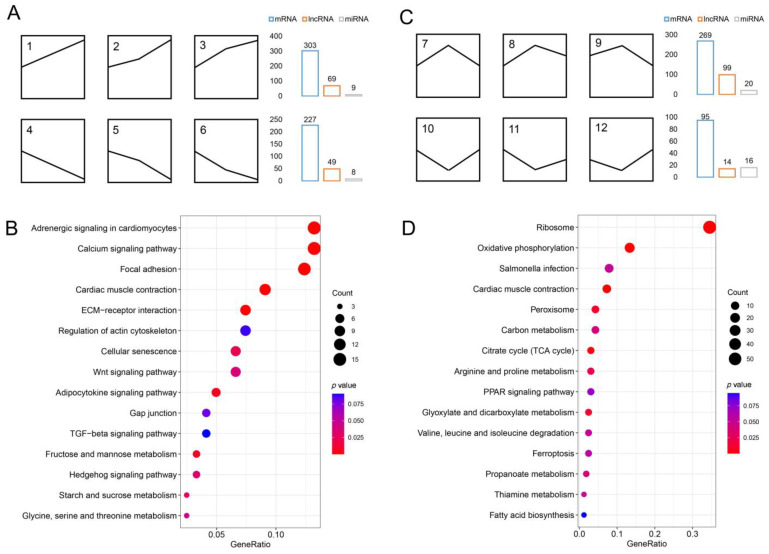
Identification of mRNA, lncRNA, and miRNA associated with the development and formation of IBs. (**A**) Six gene expression patterns were associated with IB development: 1, 2, and 3, were upregulated, and 4, 5, and 6 were downregulated. (**B**) The top 15 pathways from KEGG enrichment results of DEmRNAs were associated with the growth of IBs. (**C**) Six gene expression patterns were associated with the formation of IBs: 7, 8, and 9, were upregulated, whereas 10, 11, and 12 were down-regulated. (**D**) The top 15 pathways from KEGG enrichment results of DEmRNAs were associated with the formation of IBs.

**Figure 4 biology-12-00075-f004:**
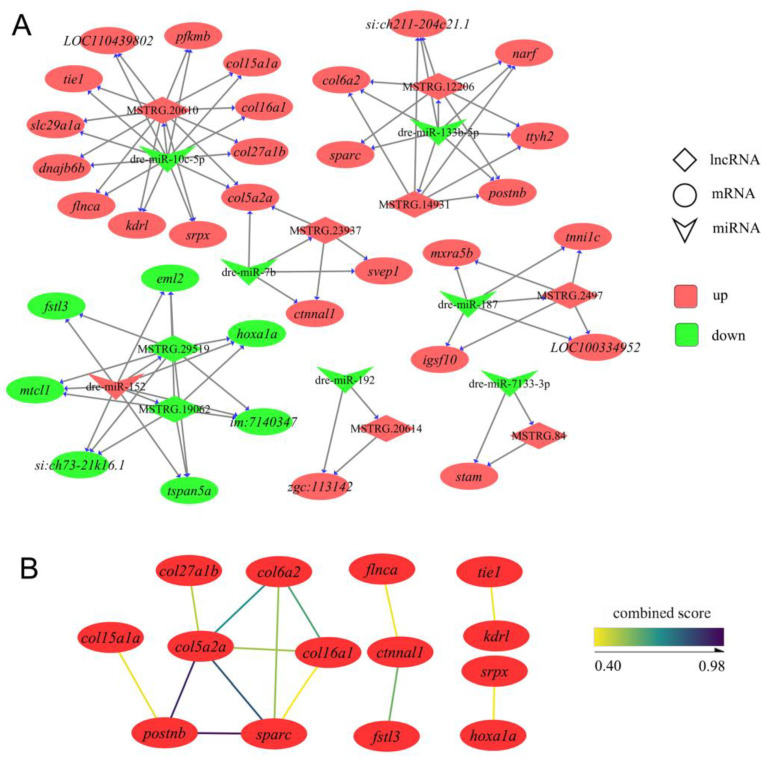
The details of ceRNA networks associated with the development of IBs. (**A**) The ceRNA networks are associated with IB development. Red represents upregulation, and the green represents downregulation. (**B**) Protein interactions are present in the ceRNA networks.

**Figure 5 biology-12-00075-f005:**
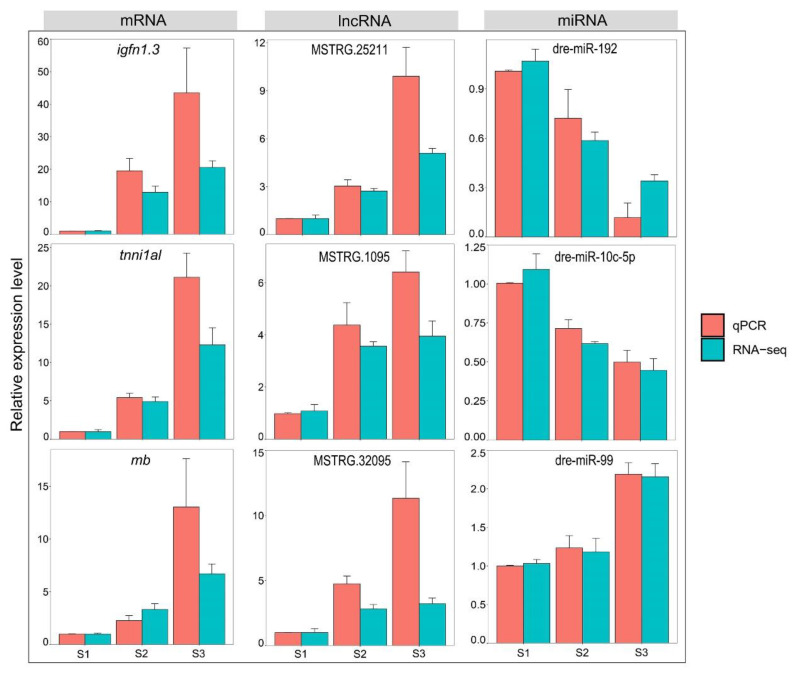
Comparison of RNA-seq data and RT-qPCR data.

**Table 1 biology-12-00075-t001:** The DEmRNAs in known pathways associated with bone development.

Pathway	Upregulated Genes	Downregulated Genes
ECM–receptor interactions	*col6a1*, col6a2, *itga11b*, thbs1a, *thbs3a*, *thbs4a*, *itga9*	*col6a4a*, *lama5*
Calcium signaling pathway	*adrb2a*, *pde1ca*, *cacna1sa*, *pdgfbb*, *plce1*, *kdrl*, *pdgfra*, *ryr1a*, *atp2a2a*, *slc25a5*, *si:dkey-163m14.2*, *tnnc1b*, *zgc:56235*	*slc8a4b*, *calm3a*, *ppp3r1b*
Hedgehog signaling pathway	*cdon*, *hhatlb*	*lrp2a*, *csnk1da*
Wnt signaling pathway	*wnt10b*, *rspo3*, *sfrp2*	*ctnnbip1*, *sfrp5*, *mycb*, *ppp3r1b*, *smad3b*
TGF-β signaling pathway	*inhbaa*, *thbs1a*, *bmpr1bb*	*mycb*, *smad3b*

## Data Availability

All clean data files are publicly available in the BIG database under the GSA accession number CRA007118.

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
