# Peer review of "Profiling the Spatial Expression Pattern and ceRNA Network of lncRNA, miRNA, and mRNA Associated with the Development of Intermuscular Bones in Zebrafish"

_biology, 2022, doi:10.3390/biology12010075_

Round 1
Reviewer 1 Report
Ye et al. studied the development of intermuscular bone in zebrafish. The authors investigated the co-expression networks of microRNAs, protein-coding RNAs and long non-coding RNAs, and identified hubs of co-expression and competitive endogenous RNA (ceRNA) networks across mRNAs/lncRNAs and miRNAs. While the results are interesting and can provide meaningful information about intermuscular bone development, there are some lacunae/uncertainties that need to be clarified.
1. The authors seem to have only considered the positive effects of lncRNAs, but lncRNAs are also known to inhibit gene expression. For ceRNA networks associated with IB development, the authors chose only positive correlations between mRNAs and lncRNAs (>=0.7), why not consider negative correlations? For example, some mRNAs inhibit the development of IBs, while they themselves are inhibited by some lncRNAs?
2. The author did not disclose family information of the zebrafish used in this study. Did all the zebrafish come from a single family? Without enough biological replicates, genetic differences across different lineages can have a greater impact on outcomes than treatments.
3. The authors described the gene expression trends over three periods and used nine samples. So, how did samples match time periods? Did the authors take the mean expression of three samples at each stage for trend analysis? This should be explained in more detail.
4. The DEGs identified by the author were significantly enriched in many pathways. So, what distinct pathways do this study find compared to previous studies? And what special significance might these pathways have?
5. Why choose 30 dpf zebrafish? Are the intermuscular bones of 30 dpf zebrafish in the same family at the same developmental stage? For zebrafish at the same stage, if there is a significant difference in body length, do they have the same state of intermuscular bone development?
6. Are the three tissue segments of a fish closely adjacent or separated?
7. The authors identified mRNA, lncRNA and miRNA associated with the formation and development of intermuscular bones. However, why did the authors construct only ceRNA networks associated with intermuscular bone development, but no with intermuscular bone formation?
8. When processing miRNA data, were sequences that were too long or too short removed? In fact, this is a very important step, but I didn't see the authors mention it. This should be explained.
9. How did the authors enrich the functions of DElncRNAs and DEmiRNAs? This should be described in the method section.
10. The extra period in line 167 should be removed.
Author Response
Response to Reviewer 1 Comments
Point 1: The authors seem to have only considered the positive effects of lncRNAs, but lncRNAs are also known to inhibit gene expression. For ceRNA networks associated with IB development, the authors chose only positive correlations between mRNAs and lncRNAs (>=0.7), why not consider negative correlations? For example, some mRNAs inhibit the development of IBs, while they themselves are inhibited by some lncRNAs?
Response 1: There is no doubt that lncRNA has many ways to regulate mRNA, and both positive and negative correlations between them are possible. However, this study focuses on ceRNA network, which is a specific model in which miRNA is the hub RNA while mRNA and lncRNA competitively combine miRNA so as to reach a crosstalk of these three RNAs. It can be understood as a model in which lncRNA indirectly regulates mRNA expression through microRNA, and it is just one of many lncRNA-mRNA regulation models. In this mode, upregulated lncRNA will bind more miRNAs and suppressed the target mRNA's degradation or inhibition induced by miRNA, which is ultimately showed as the upregulation of mRNA expression, and vice versa. Therefore, the expression levels of mRNA and lncRNA in the ceRNA network show the same trend and a positive correlation. Thus, we also screened mRNA and lncRNA pairs based on this principle. In addition to the correlation between the expression trends of mRNA and lncRNA, ceRNA network analysis also takes into consideration whether mRNA and lncRNA are the target genes of the same miRNA, and the results should be more reliable.
Point 2: The author did not disclose family information of the zebrafish used in this study. Did all the zebrafish come from a single family? Without enough biological replicates, genetic differences across different lineages can have a greater impact on outcomes than treatments.
Response 2: We strongly agree with the reviewer's opinion. In order to reduce the noise caused by inconsistent genetic background, we selected zebrafish from the same family for sampling. We have added these descriptions in section 2.2 (“Sample collection and RNA extraction”) of the manuscript.
Point 3: The authors described the gene expression trends over three periods and used nine samples. So, how did samples match time periods? Did the authors take the mean expression of three samples at each stage for trend analysis? This should be explained in more detail.
Response 3: 30dpf zebrafish with the IBs growing to the mid-lower dorsal fin were used for sampling, including the S1 segment (IBs have not yet formed), the S2 segment (few IBs have been formed), and the S3 segment (all IBs have been formed). We take the mean expression of three samples at each stage for trend analysis. We have added these descriptions in section 2.5 (“Differential expression, gene expression pattern, and function enrichment analysis“) of the manuscript.
Point 4: The DEGs identified by the author were significantly enriched in many pathways. So, what distinct pathways do this study find compared to previous studies? And what special significance might these pathways have?
Response 4: In fact, the pathways we found have been reported in previous studies, but the ones we found are more comprehensive and prominent.
Point 5: Why choose 30 dpf zebrafish? Are the intermuscular bones of 30 dpf zebrafish in the same family at the same developmental stage? For zebrafish at the same stage, if there is a significant difference in body length, do they have the same state of intermuscular bone development?
Response 5: Since the IBs of the 30 dpf zebrafish grow to the middle and lower part of the dorsal fin, it is easy for us to cut the IBs development segment more evenly. The developmental state of the IBs of 30dpf zebrafish in the same family is different. Therefore, we observed the developmental state of IBs of each zebrafish before sampling, and selected zebrafish at the same developmental state for sampling. The developmental state of IBs of zebrafish varies significantly with different body length. We have added these descriptions in section 2.2 (“Sample collection and RNA extraction”) of the manuscript.
Point 6: Are the three tissue segments of a fish closely adjacent or separated?
Response 6: The three tissue segments of a fish are closely adjacent, there are no intervals between the three segments sampled. We have supplemented descriptions for this in section 2.2 (“Sample collection and RNA extraction”) of the manuscript.
Point 7: The authors identified mRNA, lncRNA and miRNA associated with the formation and development of intermuscular bones. However, why did the authors construct only ceRNA networks associated with intermuscular bone development, but no with intermuscular bone formation?
Response 7: In fact, we constructed the ceRNA networks for IBs formation, but we did not get useful information from it, so we deleted it.
Point 8: When processing miRNA data, were sequences that were too long or too short removed? In fact, this is a very important step, but I didn't see the authors mention it. This should be explained.
Response 8: When processing miRNA data, we deleted sequences greater than 31nt and less than 18nt. We have added these descriptions in section 2.4 (“Assembly and annotation of transcriptomes”) of the manuscript.
Point 9: How did the authors enrich the functions of DElncRNAs and DEmiRNAs? This should be described in the method section.
Response 9: The essence of functional enrichment of miRNA and lncRNA is functional enrichment of their target genes. The target genes of miRNA are the mRNAs that have a targeting relationship with them, and their expression are highly correlated. The target genes of lncRNA are mRNAs with high correlation with their expression. We have added these descriptions in section 2.5 (“Differential expression, gene expression pattern, and function enrichment analysis“) of the manuscript.
Point 10: The extra period in line 167 should be removed.
Response 10: We have modified it according to the reviewer's suggestion.
Reviewer 2 Report
The manuscript of Weidong Ye et al describes gene expression signatures associated with growth of intermuscular bones (IBs) in zebrafish. The formation of these bones is relatively rarely studied and the manuscript has potential to provide useful information to researchers working on bone formation. In general, the manuscript is of good quality. Nevertheless, I have some comments on the manuscript.
Specific comments:
1) There is some limitations in the description of methods. Please, clarify the sample collection protocol. Does the sample contain IBs, vertebra and or muscles? This is vital in order to interpret gene expression data properly. If the sample consists of predominantly of bones, the conclusions are valid. If there is also other tissues involved, this should be clearly stated in discussion.
2) The discussion is slightly incomplete. Some of the key regulated pathways (such as focal adhesion) are not discussed, although they could be relevant.
3) Typo in Figure 4B scale bar.
Author Response
Response to Reviewer 2 Comments
Point 1: There is some limitations in the description of methods. Please, clarify the sample collection protocol. Does the sample contain IBs, vertebra and or muscles? This is vital in order to interpret gene expression data properly. If the sample consists of predominantly of bones, the conclusions are valid. If there is also other tissues involved, this should be clearly stated in discussion.
Response 1: We strongly agree with the reviewer's opinion. Our sample mainly contains the IBs and its closely connected muscles, excluding skin, vertebra, blood and excess muscle. We have added these descriptions in section 2.2 (“Sample collection and RNA extraction”) of the manuscript. We tried to ensure that the differences in gene expression between the different IBs development segments were originated from the different IBs developmental states.
Point 2: The discussion is slightly incomplete. Some of the key regulated pathways (such as focal adhesion) are not discussed, although they could be relevant.
Response 2: We have added discussions of these key pathways.
Point 3: Typo in Figure 4B scale bar.
Response 3: Thanks for the reviewer's reminding. The mistake has already been corrected.